# Acute Pediatric Chagas Disease in Antioquia, Colombia: A Geographic Location of Suspected Oral Transmission

**DOI:** 10.3390/microorganisms10010008

**Published:** 2021-12-22

**Authors:** Lídia Gual-Gonzalez, Catalina Arango-Ferreira, Laura Camila Lopera-Restrepo, Omar Cantillo-Barraza, Daniela Velásquez Marín, Natalia Restrepo Bustamante, Omar Triana-Chavez, Melissa S. Nolan

**Affiliations:** 1Arnold School of Public Health, University of South Carolina, Columbia, SC 29208, USA; 2Departamento de Pediatría Hospital San Vicente Fundación, Medellin 050010, Colombia; catalina.arango@sanvicentefundacion.com; 3Departamento de Pediatría, Facultad de Medicina, Universidad de Antioquia, MMedellin 050010, Colombia; daniela.velasquez5@udea.edu.co (D.V.M.); natalia.restrepob@udea.edu.co (N.R.B.); 4Grupo de Gastrohepatología, Facultad de Medicina, Universidad de Antioquia, Medellin 050010, Colombia; lcamila.lopera@udea.edu.co; 5Grupo Biología y Control de Enfermedades Infecciosas, Universidad de Antioquia, Medellin 050010, Colombia; omarcantillo@gmail.com (O.C.-B.); omar.triana@udea.edu.co (O.T.-C.)

**Keywords:** *Trypanosoma cruzi*, oral transmission, pediatric Chagas disease, Colombia, Antioquia

## Abstract

Chagas disease, *Trypanosoma cruzi* infection, is an insidious cause of heart failure in Latin America. Early diagnosis and treatment are critical to prevent irreversible myocardial damage that progressively accumulates over decades. Several structural barriers account for the less than 1% of cases in Colombia being treated, including poor physician knowledge, especially considering that some regions are considered non-endemic. The two cases reported here represent an emerging epidemiologic scenario associated with pediatric Chagas disease. Both cases are suspected oral transmitted parasitic infection in a geographic region of Colombia (Andean region of Antioquia) where no previous oral transmission of Chagas disease had been reported. Their clinical histories and course of disease are presented here to increase physician awareness of the epidemiologic risk factors and clinical manifestations associated with pediatric oral Chagas disease in Antioquia department, Colombia.

## 1. Introduction

Chagas disease is a parasitic infection that if untreated can lead to insidious organomegaly in a subset of chronically infected patients. This neglected tropical disease affects 7 million Latin Americans and another 75 million live at risk of acquiring incident disease [1]. Vector-borne transmission is the most common infection route; however, oral transmission via contaminated beverages or food products is becoming a greater public health concern [2]. Further, infected patients that acquire disease orally present with greater acute disease symptomology and occur in communal outbreaks [3,4,5]. Children are particularly vulnerable to severe disease and death from orally transmitted Chagas disease [6,7,8]. The geographic location of triatomine species associated with food/beverage product is largely unknown but has been reported in the Amazonian region of Venezuela-Brazil, Peru, and Colombia [2]. The identification of new geographic areas of oral transmission risk is important to both provide prompt treatment to those affected and perform interventions to prevent continued transmission to other local residents. The two case reports presented here highlight the clinical manifestations and epidemiologic risk factors for two children suspected of oral *Trypanosoma cruzi* infection in a new geographic area in Colombia.

## 2. Case Presentations

### 2.1. Case 1

On 27 May 2019, a 10-year-old previously healthy male from a split home in Medellín, Antioquia (mother), and Doradal, Antioquia (father), presented to the emergency room at the San Vicente Fundación Hospital in Medellín. His past medical history was unremarkable for allergies, surgeries, and comorbidities. On history and presenting illness evaluation, it was noted that this child’s symptoms started on May 9 with persisting recurrent fever (39–39.8 °C), asthenia, adynamia, myalgias, paleness, hyporexia, occasional diarrhea, and macular rash on the upper limbs, face, and trunk lasting approximately five days. Previous medications indicated by his mother included acetaminophen, dipyrone, and gentamicin (80 mg once a day for 5 days). After a week, the patient was taken to an outpatient clinic and sultamicillin (325 mg two times daily for five days) and metronidazole (500 mg two times daily for three days) were prescribed. On 18 May 2019, due to the persistence of symptoms, the patient went to the El Carmen local hospital. There, blood tests were negative for anemia, leukopenia, liver impairment, and renal impairment. At that moment, there was no clear diagnosis, and he was discharged. Due to persistence of symptoms, he was medicated by his father with azithromycin from 22–26 May. The patient’s father, stepmother, and stepbrother also reported having similar symptoms.

At time of presentation to our hospital on 27 May, it was noted that the patient’s parents were divorced, and the custody arrangement was that he lived with his mother in Medellin, and visited his father in the rural area of Alto del Pollo, Doradal, Antioquia, where no appropriate aqueduct or sewage system was available. He reported frequent consumption of sugar cane juice, which is frequently sold in the streets after an artisan process of trituration, and he consumed hunted wild animals. Regarding his physical examination, the patient was hemodynamically stable, with a nontoxic appearance and normal vital signs for his age, without pallor or jaundice. A few supraclavicular lymph nodes on palpation were found, which were motile, not painful, not attached to the underlying tissue, with a size under 0.5 cm in diameter. Additionally, hepatomegaly and splenomegaly were noted on palpitation. No cardiopulmonary, dermatological, or neurological findings were present.

Complete blood count (CBC) included normal leukocytes and differential. An elevated erythrocyte sedimentation rate (34 mm/h), slightly elevated C reactive protein (2.2 mg/dL), and an elevated lactate dehydrogenase (509 U/L) were noted on blood exam. Cytomegalovirus IgM serology was negative. However, Epstein Barr virus IgG-IgM anticapside serology was positive, which was considered an incidental finding since the patient’s symptoms and his relative’s symptoms could not totally be explained by an acute simultaneous EBV infection. Therefore, additional tests were ordered, which all came back negative: a Rose Bengal test, a *Leptospira* spp. IgM ELISA test, as well as a Mantoux test with 0 mm of induration after 72 h of inoculation, and negative cultures for *Salmonella* spp. and *Shigella* spp.

On 31 May 2019, the patient was evaluated by a pediatric oncologist who performed a bone marrow aspiration due to the suspicion of a lymphoproliferative syndrome; the myelocultures (fungi, tuberculosis, and *Salmonella*) and the myelogram were negative. An abdominal ultrasound was performed with a report of hepatomegaly (14.5 cm) and splenomegaly (10.5 cm). Chest X-ray did not suggest mediastinal widening or cardiomegaly.

During his hospitalization, the patient’s stepmother was hospitalized for similar symptoms and was diagnosed with *Trypanosoma cruzi* infection. Therefore, serological tests were ordered for the patient on 6 June 2019, with positive IgG antibodies to *Trypanosoma cruzi* (value = 1.27) (Weiner Chagatest, Rosario, Argentina) and positive PCR (genotype of Discrete Typing Unit TcI) documented from blood sample. Four thick blood smears were obtained without evidence of hemoparasites. On 11 June 2019, treatment with Nifurtimox was started at a dose of 9 mg/Kg/day three times a day, according to local guidelines for the treatment of Chagas disease. On 13 June, the patient was discharged with indications for outpatient follow-up in one week. Total duration of antiparasitic treatment was 60 days according to the National Colombian Government guidelines [9].

### 2.2. Case 2

On 29 July 2021, a 5-year-old male from a rural area of Remedios, Antioquia, with a past medical history of 32 weeks gestation premature birth without any notable sequel presented at San Vicente Fundación Hospital in Medellín with a 3-day history of acute-onset clinical manifestations. Symptoms included subjective fever that remitted with acetaminophen treatment, unremitting headache of moderate intensity, diffuse abdominal pain, and multiple non-bilious emetic episodes that intensified, motivating the parents to seek medical care. Physical examination revealed a low arterial blood pressure of 108/56, with a normal heart rate, respiratory rate, and oxygen saturation. Bi-palpebral edema without any inflammatory signs suggested periorbital infection, without conjunctival infection. Patient presented with normal ear, mouth, neck, as well as heart and pulmonary evaluation.

At the time of current hospitalization, the patient’s family reported living in a self-constructed house made of wood, without access to potable water, frequent consumption and at-home preparation of hunted wild animals, including capibara or “chigüiro” and wild boars. The father denied having seen triatomine bugs inside the home. The patient’s father reported that the patient was up to date on vaccinations.

On hospital admission, *Trypanosoma*-like parasites were observed in a peripheral blood smear. An abdomen and thoracic X-ray showed cardiomegaly and hepatosplenomegaly. Elevated transaminases (AST 237 U/L, ALT 237U/L) and elevated C-reactive protein (48 mg/L) were present. Leukopenia (2800 cells/mm^3^) and mild neutropenia (1500 cells/mm^3^) were seen on CBC. Serum VDRL surface antigen for hepatitis B, total antibodies for hepatitis C, and fourth generation test for HIV infection were negative. The patient was diagnosed with a non-specific *Trypanosoma* sp. infection and was referred to a specialist.

The next day, on 30 July, the patient, now hospitalized, had repeated tests by the infectious disease physician. Symptomatic treatment was given while waiting for the second round of test results, with a plan of dipyrone 300 mg four times daily if fever or pain was present. Chest X-ray and a second abdomen X-ray revealed sustained cardiomegaly and hepatosplenomegaly. A new CBC with differential revealed slightly lower leukopenia (1600 cells/mm^3^) and neutropenia (680 cells/mm^3^) with new diagnosis of thrombocytopenia (96,000 cells/mm^3^). Tests were negative for NS1 antigen, IgM and IgG for dengue virus, hepatitis A IgM, and *Trypanosoma cruzi* IgG and IgM by indirect chemiluminescent immunoassay.

On 3 August, the PCR results for *Trypanosoma cruzi* returned positive for genotype Discrete Typing Unit Tc I, and the patient was officially diagnosed with acute Chagas disease. An EKG was normal except for sinus tachycardia, and the patient had a normal echocardiogram. The patient was started on treatment with Nifurtimox 10 mg orally every 8 h for one day, 20 mg orally every 8 h for two days, and then 40 mg three times per day (8 mg/kg/day) for 57 days. On 5 August, after resolution of fever, headache, emesis, and a normal physical examination, the patient was discharged with ambulatory follow-up in one month at an outpatient clinic.

Both clinical cases summaries are listed in Table 1. The geographic locations of the suspected transmission sources are shown in Figure 1.

## 3. Discussion

This is the first case report of suspected oral-acquired *T. cruzi* infection from the Andean region of Antioquia Department, Colombia. While detailed information on oral transmission source was unavailable, the clinical sign of bipalpebral edema and the epidemiologic profile with multiple infected family members suggest oral transmission [3]. These cases may support the evidence of emerging concern of oral Chagas disease in Colombia, where in 2015 the WHO estimated 437,960 people were at risk of infection [1]. The relevance in the first case was the presentation with prolonged fever associated with hepatosplenomegaly and non-characteristic symptoms of any specific infection that included diarrhea, asthenia, myalgias, and rash, which in a tropical country can be present in numerous infectious diseases with a wide spectrum of differential diagnosis. The fact that more members of the family were also sick suggests oral transmission and presents an additional clinical clue for consideration of acute Chagas disease as a plausible diagnosis. The second case demonstrated rapid onset of clinical disease with no detection of antibodies, again suggesting an acute infection.

In cases of oral transmission of Chagas disease, bipalpebral and facial edema as well as prolonged fever appear as the most common symptoms, being present in 90–100% of cases [3]. Myalgia, headache, and hepatosplenomegaly are also common in more severe disease cases [2]. Remarkably, the rash can be present in up to 27% of cases, as was presented in case 1 [11]. Clear gastrointestinal involvement with hepatosplenomegaly can be observed in both cases [12]. The transmission mechanism is of public health interest to help prevent infections of other household members and close community members, highlighting the value of prompt differential diagnosis for a subsequent treatment.

In Colombia, the national efforts to prevent vectorial transmission are specially focused on eliminating the intradomicile vector *Rhodnius prolixus*. This national plan has been largely effective, and only a few departments remain with a burden of acute vector-borne Chagas deaths [13]. The department of Antioquia is considered to have eradicated *R. prolixus*, but secondary vectors are now thought to play an important role in continued transmission. *Panstrongylus geniculatus* is a reported secondary vector in Antioquia [14], although *Triatoma dimidiata*, *Triatoma venosa*, and *Panstrongylus rufotuberculatus* have been reported in the neighboring Boyacá department [15]. *P. geniculatus* has been described as a prominent vector associated with oral transmission via fecal contamination of juice in one’s home [11,16]. Given the commonality of this species in Antioquia, this species could have been present in the domestic environment of our two reported cases of Chagas; however, in case 1, the habit of consuming hunting meat could lead to potential oral transmission as well. This highlights the need for future studies in this region to clarify this epidemiological situation.

Oral outbreaks have been increasing in frequency in the Eastern region of Colombia; the states bordering Venezuela. A recent study of two outbreaks in 2016 confirmed 54 cases of suspected oral transmission [17]. Oral outbreaks in this region are hypothesized to occur in epidemiologic scenarios where sylvatic-domestic triatomine species overlap [11,18]. The department of Casanare has particularly suffered from oral *T. cruzi* transmission outbreaks [13,19]. Acute oral transmission could potentially be underrepresented since the parasite source is unknown in some acute Chagas disease cases [19,20]. Acute oral-acquired Chagas disease is of particular concern given the pathologic severity and mortality risk [2,21,22,23].

Despite being an endemic disease in many Latin American countries, the diagnosis of Chagas disease is far from timely, with known treatment delays [24]. In Colombia, the prevalence of Chagas disease is estimated to be between 1.67% and 2% of the population; however, it has been estimated that as few as 1.5% of cases have been diagnosed [25,26]. These two cases are the reflection of a growing problem in Colombia, and Colombia has recently become one of the Latin American countries with a considerable number of acute outbreaks of Chagas disease where oral transmission of *T. cruzi* has been recorded [27]. 

An opportune diagnosis of this disease is of worldwide concern. Olivera et al. describe three main barriers that prevent timely access to health services by patients with Chagas disease. Two of these barriers are closely related: limited diagnosis of Chagas disease and limited awareness and knowledge of physicians about the disease [28]. Many of the countries where this disease is endemic do not have enough trained laboratories for the diagnosis of Chagas or they are located in large cities, often far from rural areas [29]. It is notable, especially in case 1, that the initial suspicion was not Chagas disease and multiple tests were necessary to rule out differential diagnoses until finally reaching the correct one. It has been estimated that a lack of early diagnosis and treatment results in an annual cost of $175 million USD to the Colombian government [24]. Healthcare costs are an important limitation to Chagas disease diagnosis and treatment in Colombia. At-risk populations typically live in rural areas, with limited resources, and must travel to larger cities to obtain a diagnosis and treatment [12,24]. The two cases described here represent the common scenario of Chagas disease in Colombia, with families living in remote rural areas. These social, technical, and educational problems are the explanation for why not as many cases are diagnosed as estimated [29,30]. 

## 4. Conclusions

In pediatric populations, the control of congenital and vectorial *T. cruzi* transmission has improved. Unfortunately, as our case report indicates, oral infection is emerging as an epidemiologically important form of transmission in some areas [31], hindering the diagnosis in cases where no vector is found or no history of Chagas is known. It also has a complicated epidemiologic scenario hindered by the appearance of new vectors and low physician knowledge [15,28,32,33]. The fact that oral transmission has been identified in a new geographic region, Antioquia Department, is of importance for travel medicine physicians as well. Prior to the recent pandemic, tourism has consistently increased annually in Colombia, with an estimated half a million tourists visiting Antioquia Department annually. 

Children are a particularly vulnerable population to oral transmission [7,8], which is concerning given the higher symptomology rate associated with this transmission route [21,23,34]. Lastly, despite the large-scale Pan American Health Organization vector control efforts of the 1990s and 2000s, children are still being infected with Chagas disease at unacceptable levels [35,36]. This study highlights the importance of detection, which is especially important in children, when cardiac symptoms have been seen to develop early [37,38]. Timely treatment is crucial to prevent chronicity and early mortality [39].

## Figures and Tables

**Figure 1 microorganisms-10-00008-f001:**
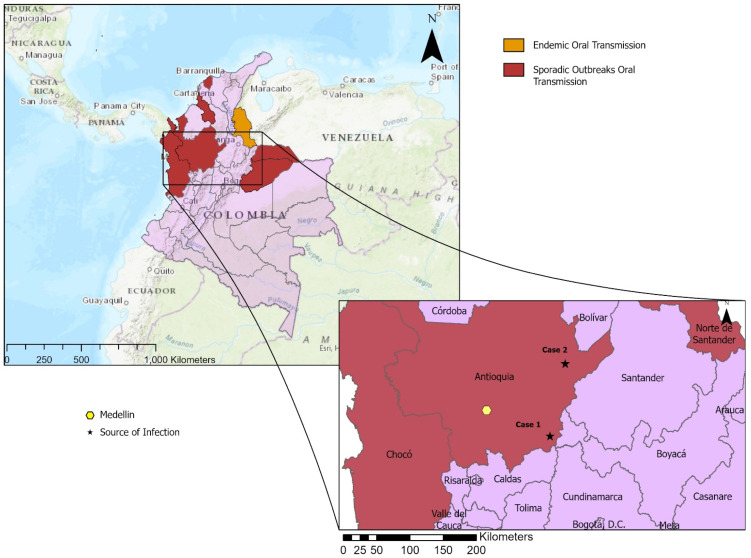
Map depicting the geographic location of cases and areas of historically endemic and sporadic outbreaks of oral transmission [10]. Map showing the locations of the possible transmission sites (stars) of the two cases: Case 1 in Alto del Pollo, Doradal, and Case 2 in Vereda Pueblo Nuevo, Remedios, both in the Department of Antioquia (Andean Region) and showing the biggest city nearby, Medellín (yellow hexagon).

**Table 1 microorganisms-10-00008-t001:** Chagas disease clinical profiles from two acute cases in Antioquia Department, Colombia, diagnosed in 2019 and 2021.

Clinical Presentations of Chagas Disease	Case 1	Case 2
Age	10 years old	5 years old
Suspected place of infection	Doradal (Antioquia)	Remedios (Antioquia)
**Acute phase**		
Fever	Present	Present
Malaise	Present	Present
Myalgia	Present	Absent
Headache	Absent	Present
Adynamia	Present	Absent
Gastrointestinal involvement	Present	Present
Hyporexia	Present	Present
Lymphadenomegaly	Present	Absent
Romaña sign	Absent	Absent
Chagoma	Absent	Absent
ECG abnormalities	Absent	Absent
Myocarditis	Absent	Absent
Leukopenia	Absent	Present
Blood abnormalities ^1^	Present	Absent
Rash	Present	Absent
Hepatosplenomegaly ^2^	Present	Present
Liver impairment ^2^	Absent	Present
**Intermediate phase**		
Heart conduction abnormalities	Absent	Absent
Regional LV wall motion abnormalities	Absent	Absent
**Chronic phase**		
Cardiomegaly	Absent	Present
Megacolon	Absent	Absent
Megaesophagus	Absent	Absent
**Blood detection**		
*Trypanosoma cruzi* observation	Not observed	Observed
Antibodies	IgG positive	Negative
PCR detection	Positive	Positive

^1^ Blood abnormalities including high levels of lactate dehydrogenase (LDH) and elevated erythrocyte sedimentation rate (ESR) ^2^ Overlapping symptoms that may occur in both acute and chronic phase.

## Data Availability

The clinical details without patient identifying information are available upon request to Melissa Nolan: msnolan@mailbox.sc.edu.

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
