# Peer review of "Acute Pediatric Chagas Disease in Antioquia, Colombia: A Geographic Location of Suspected Oral Transmission"

_microorganisms, 2021, doi:10.3390/microorganisms10010008_

Round 1

Reviewer 1 Report

  • Title: Acute pediatric Chagas disease in Antioquia, Colombia: A geo
  • graphic location of emerging oral transmission
  • I suggest: Acute pediatric Chagas disease in Antioquia, Colombia: A geographic location of oral transmission
  • Lines 63, 64: CBC included normal leukocytes and differential (L 5700, neutrophils 40% lympho- 63 cytes 53%, monocytes 3,4%, eosinophils 1,6%), with Hb of 11,8gr/dL, Hematocrit of 34% 64 MCV 72,
  • I suggest: As this is the first time the abbreviations CBC and MCV appear in the text, I suggest writing it out in full and not in abbreviated form.
  • Lines 81-89: During his hospitalization, the patient’s stepmother, who was hospitalized for simi lar symptoms, was diagnosed with Trypanosoma cruzi infection. Therefore, serological tests were ordered for the patient on June 6, 2012, with positive IgG antibodies to Trypanosoma cruzi (value=1.27) (Weiner Chagatest, Rosario, Argentina) and positive PCR (gen otype of sylvatic TcI) from blood sample were documented. Four thick blood smears were obtained without evidence of hemoparasites. On June 11, 2019, treatment with Nifurtimox  was started at a dose of 9 mg/Kg/day three times a day, according to local guidelines for  the treatment of Chagas disease. On June 13, he was discharged with indications for out- patient follow-up in one week. Total duration of antiparasitic treatment was 60 days according to the National Colombian Government guidelines [1].

I suggest: The copied text is not clear. Need clarification, as the text is written, it gives rise to several doubts. When was the 10-year-old boy's mother diagnosed with Chagas disease? Was it in 2012? In May 2019 the boy sought medical care, however the diagnosis for Chagas disease was made after ruling out several hypotheses. How was the hypothesis of congenital transmission to the 10-year-old boy discarded? Was the 10-year-old boy's stepmother diagnosed with Chagas disease on 6/6/2012? By what methodology? What tests was the boy diagnosed as having Chagas disease?

  • Line 174 : Given the commonality of this species in Antioquia,  w this species could have
  • I suggest: Given the commonality of this species in Antioquia, this species could have
  • Line 184, 185: in some acute chagas disease cases [12, 13]. Acute oral-acquired chagas disease is of particular concern given the pathologic severity and mortality
  • I suggest: in some acute Chagas disease cases [12, 13]. Acute oral-acquired Chagas disease is of particular concern given the pathologic severity and mortality

Author Response

Response to Reviewer 1 Comments

Title: Acute pediatric Chagas disease in Antioquia, Colombia: A geographic location of emerging oral transmission

I suggest: Acute pediatric Chagas disease in Antioquia, Colombia: A geographic location of oral transmission

Author response:  Thank you for your suggestion. We agree the title change provides a simpler description of the cases. Another reviewer suggested a change in the title and we proceeded to a new title as follows: “Acute pediatric Chagas disease in Antioquia, Colombia: A geographic location of suspected oral transmission

Lines 63, 64: CBC included normal leukocytes and differential (L 5700, neutrophils 40% lympho- 63 cytes 53%, monocytes 3,4%, eosinophils 1,6%), with Hb of 11,8gr/dL, Hematocrit of 34% 64 MCV 72,

I suggest: As this is the first time the abbreviations CBC and MCV appear in the text, I suggest writing it out in full and not in abbreviated form.

Author response: Thank you for your suggestion. We added the changes to the manuscript in lines 82 and 84. It reads: “Complete blood count (CBC) included normal leukocytes and differential (L 5700, neutrophils 40% lymphocytes 53%, monocytes 3,4%, eosinophils 1,6%), with Hb of 11,8gr/dL, Hematocrit of 34% Mean corpuscular volume (MCV)”.

Lines 81-89: During his hospitalization, the patient’s stepmother, who was hospitalized for simi lar symptoms, was diagnosed with Trypanosoma cruzi infection. Therefore, serological tests were ordered for the patient on June 6, 2012, with positive IgG antibodies to Trypanosoma cruzi (value=1.27) (Weiner Chagatest, Rosario, Argentina) and positive PCR (gen otype of sylvatic TcI) from blood sample were documented. Four thick blood smears were obtained without evidence of hemoparasites. On June 11, 2019, treatment with Nifurtimox  was started at a dose of 9 mg/Kg/day three times a day, according to local guidelines for  the treatment of Chagas disease. On June 13, he was discharged with indications for out- patient follow-up in one week. Total duration of antiparasitic treatment was 60 days according to the National Colombian Government guidelines [1].

I suggest: The copied text is not clear. Need clarification, as the text is written, it gives rise to several doubts. When was the 10-year-old boy's mother diagnosed with Chagas disease? Was it in 2012? In May 2019 the boy sought medical care, however the diagnosis for Chagas disease was made after ruling out several hypotheses. How was the hypothesis of congenital transmission to the 10-year-old boy discarded? Was the 10-year-old boy's stepmother diagnosed with Chagas disease on 6/6/2012? By what methodology? What tests was the boy diagnosed as having Chagas disease?

Author response: Thank you for your comment. The patient’s stepmother was hospitalized for similar symptoms during the same time the patient was hospitalized. She was diagnosed through same antibody tests in May-June 2019, (it was a typographic mistake). The stepmother’s timely diagnosis suggested the patient’s doctors the child might have contracted Chagas and thus they decided to test him. The patient’s biological mother lived in an urban area of Medellin and did not have triatomine exposures (this is stated in the first paragraph). Therefore, congenital transmission was not considered likely.

Lines 100 to 102, we clarified the text and corrected the date of the diagnosis. It reads: “During his hospitalization, the patient’s stepmother, was hospitalized for similar symptoms, and was diagnosed with Trypanosoma cruzi infection. Therefore, serological tests were ordered for the patient on June 6, 2019, with positive IgG antibodies to [..] “.

Line 174 : Given the commonality of this species in Antioquia,  w this species could have

I suggest: Given the commonality of this species in Antioquia, this species could have

Author response: Thank you for your comment. We removed the error from the sentence in line 194. It reads: “Given the commonality of this species in Antioquia, this species could have been present [..]”.

Line 184, 185: in some acute chagas disease cases [12, 13]. Acute oral-acquired chagas disease is of particular concern given the pathologic severity and mortality

I suggest: in some acute Chagas disease cases [12, 13]. Acute oral-acquired Chagas disease is of particular concern given the pathologic severity and mortality

Author response: Thank you for your comment. Line 205,206: we made the suggested changes. It reads: “ in some acute Chagas disease cases [18, 19]. Acute oral-acquired Chagas disease is of particular concern [,.]”.

Reviewer 2 Report

This article presents two cases of presumed orally acquired acute Chagas disease in Colombia. 

General comments

A succinct introduction about the general medical condition that is discussed in these case reports is missing.

Case presentations should be summarized, as there is unnecessary information that distracts from the important one. Consider to add the other acute adult cases of those outbreaks if information can be recovered.

English style should be improved, try to make more concise sentences

A conclusion briefly outlining the take-home messages and the lessons learned is missing

As the article is referred to a very specific location, I'd suggest to add a comment on implications for travel clinic physicians, for example, when they advise travellers to Colombia, so it increases interest for global readers.

Specific comments

Line 2: Change title to include that these are cases of presumed or suspected oral transmission

Line 15: explain why authors that co-contributed as first authors are not first and second authors

Line 23: in a new geographic region of Colombia (Andean region of Antioquia) should be changed for a geographic region of Colombia (Andean region of Antioquia) non endemic for CD or for a geographic region of Colombia (Andean region of Antioquia) where no previous oral transmission of CD had been reported.

Line 86 and 131: explain why nifurtimox is used instead of benznidazole.

Table 1: change home residency for suspected place of CD acquisition. Hepatosplenomegaly and Liver impairment can occur in the acute phase and should be placed there as the cases presented are acute cases. Mega-esophagous or megacolon are chronic manifestations

Figure 1: if possible, mark areas endemic for Chagas disease

Line 145-146: The sentence “While detailed information on oral transmission source was unavailable, the clinical and epidemiologic profiles suggest oral transmission” should be explained, describing the clinical and epidemiological profile of oral transmission that helps differentiate from vectoral transmission. I’d suggest including this reference: https://doi.org/10.1016/j.tmaid.2020.101565.

Line 160: add a comment on bipalpebral oedema as very typical of orally acquired CD

Line 179: Please add information on Colombia acute CD epidemiology (number of cases reported, number of outbreaks, regions with cases reported)

Line 208: please remove “without adequate hygiene and sanitation to prevent parasite transmission” as CD transmission is associated with poverty but no inadequate sanitation.

Line 211: please provide some information on congenital CD in Colombia

Line 236: please change references to the journal format

(Journal Articles: 1. Author 1, A.B.; Author 2, C.D. Title of the article. Abbreviated Journal Name YearVolume, page range.)

Author Response

Response to Reviewer 2 Comments

This article presents two cases of presumed orally acquired acute Chagas disease in Colombia. 

General comments

A succinct introduction about the general medical condition that is discussed in these case reports is missing.

Author response : Thank you for your comment. We have added a succinct introduction to the issue and the role the current case series in addressing the knowledge gap.

Case presentations should be summarized, as there is unnecessary information that distracts from the important one. Consider to add the other acute adult cases of those outbreaks if information can be recovered.

Author response: Thank you for your comment. After much discussion between the authorship team (including the original clinicians), we would request approval for the clinical information provided in the current format to be kept in. We greatly reduced each case’s history and presenting illness and clinical course of disease, as is. Each patient was in the hospital for a considerable amount of time, and the information currently listed is thought to be succinct and each detail of clinical relevance.

English style should be improved, try to make more concise sentences

Author response: We have revised the wording throughout to ensure verbal flow and grammatical correctness.

A conclusion briefly outlining the take-home messages and the lessons learned is missing

As the article is referred to a very specific location, I'd suggest to add a comment on implications for travel clinic physicians, for example, when they advise travellers to Colombia, so it increases interest for global readers.

Author response: We have added a conclusion section to the article, and have added why our findings might have relevance for travel medicine clinicians.

Specific comments

Line 2: Change title to include that these are cases of presumed or suspected oral transmission.

Author response: Thank you for your suggestion. We made a change in the title. It reads: “Acute pediatric Chagas disease in Antioquia, Colombia: A geographic location of suspected oral transmission”

Line 15: explain why authors that co-contributed as first authors are not first and second authors

Author response: Thank you for your comment. Two authors co-contributed as first authors on the preparation of the manuscript, while the second author contributed in the investigation and management of the cases. We considered and agreed on making the second author one of the doctors that attended the cases.

Line 23: in a new geographic region of Colombia (Andean region of Antioquia) should be changed for a geographic region of Colombia (Andean region of Antioquia) non endemic for CD or for a geographic region of Colombia (Andean region of Antioquia) where no previous oral transmission of CD had been reported.

Author response: Thank you for your suggestion. We agree the change in the sentence provides more clear explanation of the geographical distribution of the cases. We made the suggested changes in line 68,69. It reads: “a geographic region of Colombia (Andean region of Antioquia) where no previous oral transmission of Chagas disease had been reported.”

Line 86 and 131: explain why nifurtimox is used instead of benznidazole.

Author response: Thank you for your comment.  Nifurtimox is currently the most commonly used medication in Central America and the northern region of South America. This is likely due to Nifurtimox being donated by its manufacturer to those governments and the recent publications on nifurtimox formulation for pediatric populations.

Table 1: change home residency for suspected place of CD acquisition. Hepatosplenomegaly and Liver impairment can occur in the acute phase and should be placed there as the cases presented are acute cases. Mega-esophagous or megacolon are chronic manifestations

Author response: Thank you for your suggestion. We agree to rather provide the information of the suspected site in the table as provided in the figure. We made the changes on the table as suggested. Due to overlapping symptoms between chronic and acute symptoms we decided to add a footnote in the table for Hepatosplenomegaly and Liver impairment. It reads: “Overlapping symptoms that may occur in both acute and chronic phase”.

Figure 1: if possible, mark areas endemic for Chagas disease.

Author response: Thank you for your comment. The figure is to depict the distance between the cases and show they both belong to Antioquia department, for simplicity we considered to only show where the cases occur.

Line 145-146: The sentence “While detailed information on oral transmission source was unavailable, the clinical and epidemiologic profiles suggest oral transmission” should be explained, describing the clinical and epidemiological profile of oral transmission that helps differentiate from vectoral transmission. I’d suggest including this reference: https://doi.org/10.1016/j.tmaid.2020.101565.

Author response: Thank you for your suggestion. We agree in the suggested changes to provide a more detailed explanation of why oral transmission is suspected. We made the changes in Line 165, 166, it reads:the clinical sign of bipalpebral edema and the epidemiologic profile with multiple infected family members suggest oral transmission[...]”. We added the suggested reference.

Line 160: add a comment on bipalpebral oedema as very typical of orally acquired CD.

Author response: Thank you for your suggestion. We rewrote lines 177-180 according to the suggestion. It reads: “In cases of oral transmission of Chagas disease, bipalpebral and facial edema as well as prolonged fever appear as the most common symptoms, being present in 90-100% of cases [3].Myalgia, headache, and hepatosplenomegaly are also common in more severe disease cases [2]

Line 179: Please add information on Colombia acute CD epidemiology (number of cases reported, number of outbreaks, regions with cases reported).

Author response: Thank you for your suggestion. A sentence was added with information on the recent study of cases reported in two large outbreaks in Colombia, lines 200, 201. It reads: “A recent study of two outbreaks in 2016 confirmed 54 cases of suspected oral transmission”.

Line 208: please remove “without adequate hygiene and sanitation to prevent parasite transmission” as CD transmission is associated with poverty but no inadequate sanitation.

Author response: Thank you for your suggestion. We removed the sentence as suggested.

Line 211: please provide some information on congenital CD in Colombia

Author response: Thank you for your suggestion. This article is mainly focused on the oral transmission of Chagas disease. In Colombia there is control on congenital transmission of Chagas disease and diagnosis of cases is done periodically. Successful vector control has decreased the vector transmission of the disease. To not distract from the main focus of the paper we considered it was not necessary to add detail on the congenital transmission.

Line 236: please change references to the journal format.

(Journal Articles: 1. Author 1, A.B.; Author 2, C.D. Title of the article. Abbreviated Journal Name YearVolume, page range.)

Author response: Thank you for your comment. The references have been re-formatted according to the journals format.

Round 2

Reviewer 2 Report

Although some improvements have been made, the authors have not yet made important corrections that were asked:

1. Case presentations should be summarized: unnecessary information  should be removed (such as, for example, "the patient did not have a vaccination card, although the father reported he was up to date on vaccinations"). Information should be given in a more concise and technical way.

2. Information on Colombia acute CD epidemiology (number of cases reported, number of outbreaks, regions with cases reported) should be added. The sentence added by authors on two outbreaks is not enough, Colombia national epidemiological information should be included (such as the one published in the Weekly Epidemiological Bulletins), including the number of oral transmission cases and the number of congenital transmission and vectorial transmission cases reported yearly in both adults and children.

3. Areas endemic for Chagas disease should be marked in the map presented in the figure or an additional map with those areas should be presented beside the map.

Author Response

Response Reviewer 2

  1. Case presentations should be summarized: unnecessary information  should be removed (such as, for example, "the patient did not have a vaccination card, although the father reported he was up to date on vaccinations"). Information should be given in a more concise and technical way.

Author response 1. Thank you for your comments and suggestions. The article has been shortened leaving the essential information for the complete understanding of the cases.

  1. Information on Colombia acute CD epidemiology (number of cases reported, number of outbreaks, regions with cases reported) should be added. The sentence added by authors on two outbreaks is not enough, Colombia national epidemiological information should be included (such as the one published in the Weekly Epidemiological Bulletins), including the number of oral transmission cases and the number of congenital transmission and vectorial transmission cases reported yearly in both adults and children.

Author response 2. As recommended, a more thorough review of the Chagas Disease epidemiology has been added to the discussion. Lines 209 – 216 contain the paragraph with the suggested information. It reads: “Oral outbreaks have been increasing in frequency in the Eastern region of Colombia; the states bordering Venezuela. A recent study of two outbreaks in 2016 confirmed 54 cas-es of suspected oral transmission and in 2019, six oral outbreaks were reported in Colom-bia[17,18]. The departments with the most reported acute cases are Boyacá, Santander, Norte de Santander, Arauca and Casanare, the later contributing around half the cases of acute CD on 2019 [19]. In 2021, 16 cases of acute CD have been reported, adding to the 293 cases reported between 2012-2020 [19-21]. The PAHO estimates that 1,000 cases of con-genital transmission happen every year in Colombia and around 5,200 are vector-borne, however there is insufficient data on the transmission rate and some of these cases have been recently in decrease [22,23].”

  1. Areas endemic for Chagas disease should be marked in the map presented in the figure or an additional map with those areas should be presented beside the map.

 Author response 3. Thank you for the suggestion. We considered the areas of endemic oral transmission of CD and sporadic outbreaks of oral CD to be represented in the map presented, highlighting those in a different color. The changes have been made and added in Figure 1.
